# A metamaterial-enabled design enhancing decades-old short backfire antenna technology for space applications

J. Daniel Binion[1], Erik Lier[2], Thomas H. Hand[2], Zhi Hao Jiang [1] & Douglas H. Werner[1]

Nearly two decades of intense study have passed since the term metamaterials was first introduced in 1999. In spite of their great promise, however, metamaterials have been slow to find their way into practical devices, and examples of real-world applications remain rare. In this paper, an Advanced Short Backfire Antenna (A-SBFA), augmented with anisotropic metamaterial surfaces (metasurfaces), has been designed to achieve a very high aperture efficiency across two frequency bands. This performance is unprecedented for an antenna that has seen widespread use, but few design changes over its more than 50 year existence. The reduced weight, compact design, hexagonal aperture, high dual-band efficiency, high cross-polarization isolation, as well as low multipaction and passive intermodulation (PIM) risk make the A-SBFA ideal for spaceborne applications. This transformative design demonstrates how practical metamaterials, when applied to conventional antenna technology, can provide significant performance enhancements.

[1] Department of Electrical Engineering, Computational Electromagnetics and Antennas Research Laboratory (CEARL), The Pennsylvania State University, University Park 16802 PA, USA. [2] Lockheed Martin Space, Littleton 80127 CO, USA. Correspondence and requests for materials should be addressed to D.H.W. (email: dhw@psu.edu)

The short backfire antenna (SBFA), first published in 1965[1], has been widely used in space, maritime, and terrestrial-based applications by virtue of its high gain, compact design, and rugged construction. As an example, this antenna was used in the communication link between the NASA ground station and the moon capsule of the Apollo spacecraft[2]. Composed of a cylindrical cavity containing a feed element placed between a ground plane and a smaller sub-reflector, the basic low profile design of the SBFA has seen only minor changes over the past 50 years[3–6]. Compared to end-fire antennas like the Yagi (the traditional rooftop TV antenna), it can achieve a similar gain while requiring only about one tenth of the height. Still, the antenna has a few disadvantages. First, its aperture efficiency, measured as signal strength or gain divided by its aperture area, is relatively low, typically 84% of ideal at a single frequency. Second, it does not lend itself to dual-band or broadband applications over a fractional bandwidth of more than 15%. Third, its cross-polarization isolation performance, measuring the interference between two orthogonal polarizations, is poor, hindering frequency reuse which is critical for efficient satellite communications systems. Lastly, the conventional SBFA is only efficient with a circular aperture, precluding the use of hexagonal or square apertures in antenna arrays to achieve 100% packaging efficiency. We have found that the maximum aperture efficiency achievable at a single frequency for a hexagonal SBFA is below 75%, and the highest average aperture efficiency simultaneously achievable at two frequencies is around 68%.

A new circular SBFA, which begins to overcome some of these deficiencies, was recently published[7]. The antenna is equipped with hard electromagnetic (EM) inner walls[8–10], which support a plane wave or hybrid TEM mode in the cavity, leading to near-uniform fields over the aperture. This results in high directivity and low cross-polarization compared to the conventional SBFA. Close to 100% aperture efficiency is achievable, corresponding to a directivity increase of 19% (0.75 dB) compared to a conventional antenna with the same aperture size. Simulated results of the hard-walled SBFA concept indicated that significant performance improvements to the SBFA were possible by using engineered surfaces[7]. However, these results were not validated through fabrication and measurement. Furthermore, a circular aperture offers less packaging efficiency when used in antenna arrays, and the addition of a dielectric slab within the cavity significantly increases weight, severely limiting this antenna's suitability for satellite-based applications. Fortunately, as demonstrated here, these disadvantages can be eliminated using thinner, lightweight metasurfaces.

Metasurfaces, the two-dimensional version of electromagnetic metamaterials, consist of a thin sheet of sub-wavelength, periodic artificial structures and have shown great potential for use in many diverse applications such as wave-front shaping, waveguide miniaturization, scattering control, and ultra-thin absorbers[11–16]. It was recently demonstrated that dispersion-engineered metasurfaces can be employed to line the walls of broadband horn antennas in order to make the aperture fields more symmetric by forcing the electric fields along the walls to zero[17–21]. Known as a soft boundary condition, this was previously achieved by employing transverse dielectric-filled corrugations, which add cost and weight to an antenna, as well as limit the bandwidth. Analogous to this, hard electromagnetic walls maximize the electric field along a boundary and were originally demonstrated using longitudinally oriented metal strips on top of a dielectric wall liner and longitudinally oriented dielectric-filled corrugations. Whereas traditional hard surfaces require thick dielectric slabs, this effect can also be achieved using thinner, lightweight metasurfaces. Though metamaterials are of great theoretical interest and are a significant area of current research, there are relatively few scenarios in which metamaterials have been widely adopted in practical applications.

In this work, we demonstrate a highly efficient, hexagonal, lightweight, dual-band SBFA that employs metasurfaces to control the electric field along the walls of the SBFA, creating near-uniform fields over the antenna aperture, and leading to nearly 100% aperture efficiency, similar to what occurs in a hard horn[8–10]. This unprecedented performance is achieved through the use of interior foam walls lined with anisotropic and dispersive metasurfaces that have been optimized to enhance antenna directivity at two frequency bands. In contrast to conventional hard surfaces, metasurfaces exhibit a wider range of anisotropic and dispersive properties that can be precisely engineered by known and low cost manufacturing techniques. The inclusion of metasurfaces greatly expands the design space, enabling the optimization of an SBFA that simultaneously fulfills multiple performance objectives. The metasurfaces enable hybrid modes to propagate within the SBFA cavity, allowing for the use of non-circular apertures, which are desirable for optimal array packaging. The triangular grid, achieved using hexagonal apertures, is preferred for most electronically steered array applications since it results in an approximately 13% reduction in the number of elements compared to a rectangular grid for the same directivity. This new A-SBFA represents a transformative milestone in the practical application of metamaterials technology as the first demonstration of an antenna enabled by a hard metasurface. Furthermore, the metamaterial-enabled A-SBFA presented here is uniquely suited to space-borne communication systems and has many advantages over current technology that makes it a promising candidate for the next-generation of satellite-based antenna arrays. Antennas for satellite applications need to be lightweight to reduce launch costs, and relatively small for effective packaging on board a crowded satellite payload. The high performance A-SBFA presented in this paper fulfills both of these constraints by virtue of its compact, low profile design and through the use of low-permittivity foam for reduced weight. The factor of five height difference between the A-SBFA and the helical antennas currently used for GPS communications results in significantly reduced volume for our antenna compared with current technology. In addition, satellite antennas operate in a vacuum and often at high power levels, which could cause multipaction and passive intermodulation (PIM). This risk is reduced in the A-SBFA by producing parts of the antenna via an additive manufacturing process, which has the additional benefits of reducing weight and cost.

## Results

**A-SBFA geometry**. A model of the metamaterial-enabled A-SBFA is shown in Fig. 1, where detailed dimensions can be found in Supplementary Fig. 1 and Supplementary Table 1. The height difference between the A-SBFA and the helical antennas currently used for GPS communications is visualized in Fig. 1b. The hexagonal aperture has a 19.05 cm apothem, which corresponds to a width of $2\lambda$ in the GPS L1 band (1.575 GHz). An array with a $2\lambda$ element spacing corresponds to a 14° field of view, which is ideal for medium earth orbit (MEO) applications such as GPS (Supplementary Fig. 2)[22]. The ground plane and side walls form a cavity, which serves to reflect the energy radiated from a feed antenna placed within the cavity. In contrast to conventional SBFAs, which are often fed by a dipole antenna, the feed antenna used here is a slot-loaded suspended patch, since this antenna is easier to impedance match over a wider bandwidth[6]. A circular sub-reflector, much smaller than the cavity diameter, reflects the electromagnetic (EM) energy back from the feed, setting up a

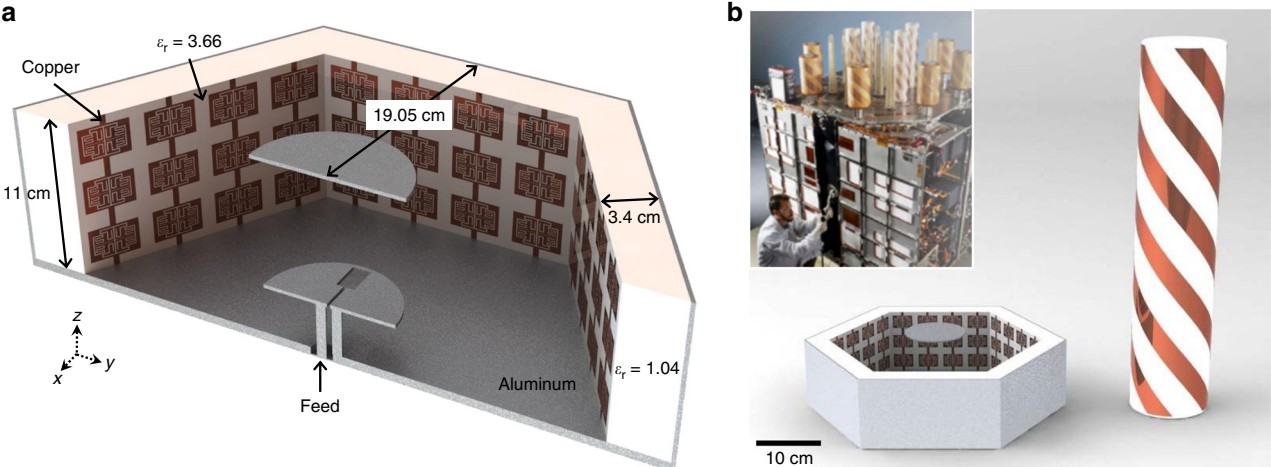

**Fig. 1** Short backfire antenna geometry. **a** Model of hexagonal metamaterial-enabled A-SBFA. **b** Visualization of size comparison between A-SBFA and helical antenna currently used aboard GPS satellite, (complete helical array shown in inset)

standing wave within the cavity. EM energy is radiated from the annular aperture between the wall and sub-reflector.

Without the metasurface lining, the SBFA has a relatively low directivity, with peak directivity occurring over a narrow bandwidth. In fact, our analysis has shown that for the hexagonal SBFA without the metasurface liner, the highest average aperture efficiency simultaneously achievable at the L1 and L2 center frequencies is around 68% (Supplementary Figs. 3–4 and Supplementary Table 2). When carefully designed, a metasurface approximates a uniform sheet impedance, which enforces a particular relationship between the electric and magnetic fields incident on the sheet, significantly altering the propagating modes within the cavity and thus, the aperture fields (Supplementary Figs. 5–7). Metasurfaces, like metamaterials, are intrinsically dispersive, which can be advantageous if a different impedance is required at different frequency bands. By tailoring this dispersive property appropriately, the metasurface can simultaneously improve peak directivity and enable dual-band operation.

**Design and optimization procedure**. Due to the complexity of the A-SBFA, there is no known method for analytically determining the metasurface impedance values required to achieve desired performance objectives. This necessitates the use of an optimization procedure. The Covariance Matrix Adaptation Evolutionary Strategy (CMA-ES) is a global optimization algorithm that has been shown to be well suited to finding solutions for a wide variety of electromagnetics problems[23,24]. In this case, CMA-ES was paired with HFSS, a commercial computational electromagnetics solver, to optimize the geometry and metasurface characteristics of the A-SBFA, with the goal of maximizing peak directivity at GPS bands L1 (1.575 GHz) and L2 (1.227 GHz). Fortunately, increased directivity naturally results in improved cross-polar isolation from a more uniform aperture excitation. At microwave frequencies metasurfaces are typically implemented with a periodic array of subwavelength unit cells composed of metallic patterns etched on a dielectric substrate. The subwavelength features require a fine mesh, which results in a computationally expensive simulation due to the large size of the domain. For this reason, the metasurfaces were modeled using homogeneous anisotropic impedance surfaces (AIS). The surface reactance values at both frequency bands of interest were included as optimization variables in addition to those representing the A-SBFA dimensions. Rather than limit the optimization to only choose surface reactance values that would approximate a

hard boundary or fulfill the balanced hybrid condition[17] we allowed the algorithm to explore the entire design space.

**Metasurface design**. Conventional hard electromagnetic surfaces are realized using dielectric-filled longitudinal corrugations or longitudinal metallic strips on a dielectric liner[8]. Such structures were found to be unsuitable for this application as they could not be designed to satisfy the required surface reactance values at both frequency bands, nor allow for a low mass foam liner. The optimized values of $X_{xx}$ and $X_{yy}$ were 200Ω and −400Ω at L2, and 89Ω and −1927Ω at L1, respectively, where $x$ and $y$ refer to a local coordinate system corresponding to the face of each of the cavity's vertical walls, with $x$ oriented in the vertical direction. The surface reactance of parallel vertically oriented metallic strips is inductive in the vertical direction and capacitive in the orthogonal direction. The reactance of such a structure can be tuned by modifying the width and spacing of the strips, but according to Foster's reactance theorem, the reactance must monotonically increase with frequency. Since both the $X_{xx}$ and $X_{yy}$ values at L1 are lower than at L2, a metasurface consisting of sub-wavelength electric resonators is required.

The metasurface unit cell, shown in Fig. 2a (with detailed dimensions shown in Supplementary Fig. 8 and Supplementary Table 3), was optimized in an infinitely periodic environment. The unit cell was illuminated with a normally incident plane wave and the effective surface reactance values of the metasurface were calculated from the simulated scattering parameters using transmission line theory (Fig. 2b; see also Supplementary Note 1 and Supplementary Fig. 9). The $x$- and $y$-oriented interdigitated metallic structures create a resonant behavior in the $X_{xx}$ and $X_{yy}$ surface reactance, respectively. From a circuit theory standpoint, since the metallic structure is interconnected in the $x$-direction, the unit cell can be viewed as an inductor in series with a parallel $LC$ circuit. Similarly, the gap between the conductive portions of the unit cells in the $y$-direction creates an effective capacitance in series with a parallel $LC$ resonator (Supplementary Fig. 10). By modifying the dimensions of this structure, the $X_{xx}$ and $X_{yy}$ resonances can be independently shifted to higher or lower frequencies in order to achieve the desired surface reactance values at L1 and L2. Although a hard boundary condition was not imposed in the optimization, when considering the anisotropic effective input impedance of the metasurface placed above a ground plane representing the cavity walls, an approximately hard boundary condition is achieved at L1 and L2 (Supplementary Fig. 11).

When the homogeneous AIS in the A-SBFA model was replaced by the optimized metasurface, the simulation results showed that the peak directivity in each band had shifted to a slightly higher frequency. This is not unexpected, as the metasurface was designed in an infinitely periodic environment, so truncating it and placing it inside the A-SBFA cavity would naturally detune it. Additionally, since the effective surface reactance of the metasurface is weakly dependent on incidence angle, the surface impedance resonances can shift in frequency due to the direction of wave propagation within the cavity. A slight adjustment of the unit cell dimensions was required to shift the peak directivity back to the desired frequencies.

**Fabrication procedure**. The realized metasurface, shown in Fig. 2c, consisted of copper traces etched on a thin dielectric substrate, and was manufactured using standard printed circuit board (PCB) fabrication techniques. The metasurface sheets were then attached to foam blocks and placed in the A-SBFA. For space-based antenna systems operating in a vacuum, electrostatic discharge (ESD) caused by charge build-up on ungrounded conductive structures has to be mitigated. Since the unit cells comprising the metasurface are interconnected in the vertical direction it was straightforward to solder them to the ground plane to eliminate the risk of ESD, while simultaneously satisfying the electromagnetic requirements.

The A-SBFA cavity was fabricated out of aluminum using additive manufacturing (3D printing), which is desirable for high-power antenna applications. Manufacturing an antenna in a single structure reduces multipaction and PIM vulnerability that will otherwise degrade signal quality in a satellite link. The slotted patch antenna feed, circular sub-reflector, as well as the two-wire transmission line used to both feed and support the patch, were machined out of aluminum (see Fig. 3a). An air-filled, square coaxial transmission line network, shown in Fig. 3b, was optimized to match the input impedance of the A-SBFA to a 50 Ω coaxial

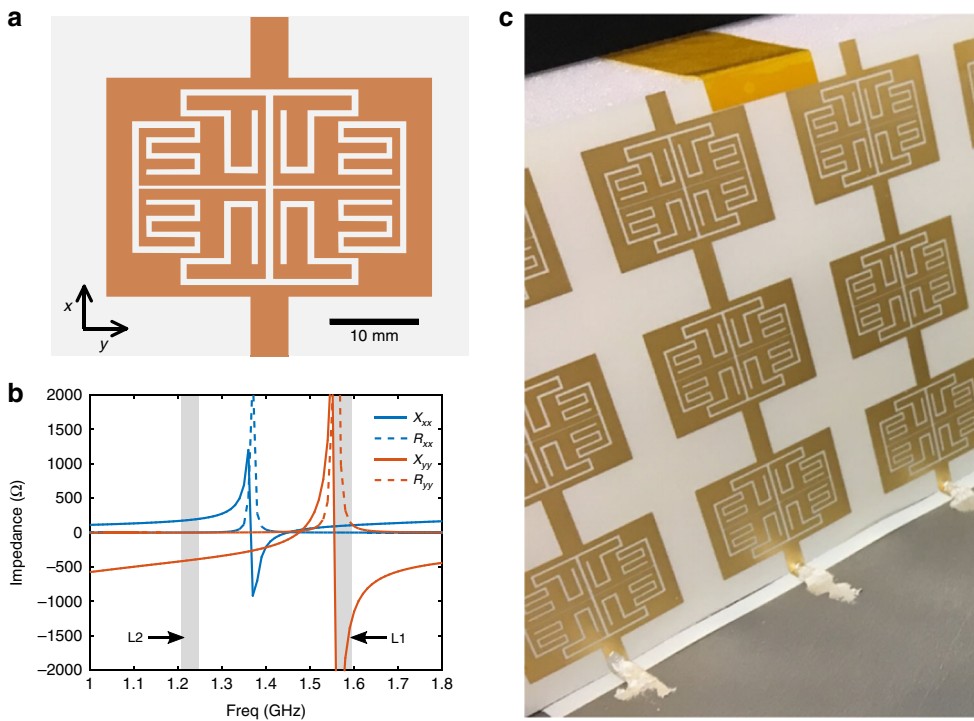

**Fig. 2** Metasurface geometry and surface impedance characteristics. **a** Geometry of a metasurface unit cell. The orange portions are copper and the white portions are thin dielectric substrate ($\epsilon_r = 3.66$). **b** Extracted real and imaginary parts of the anisotropic metasurface unit cell surface impedance. **c** A view of the fabricated metasurface

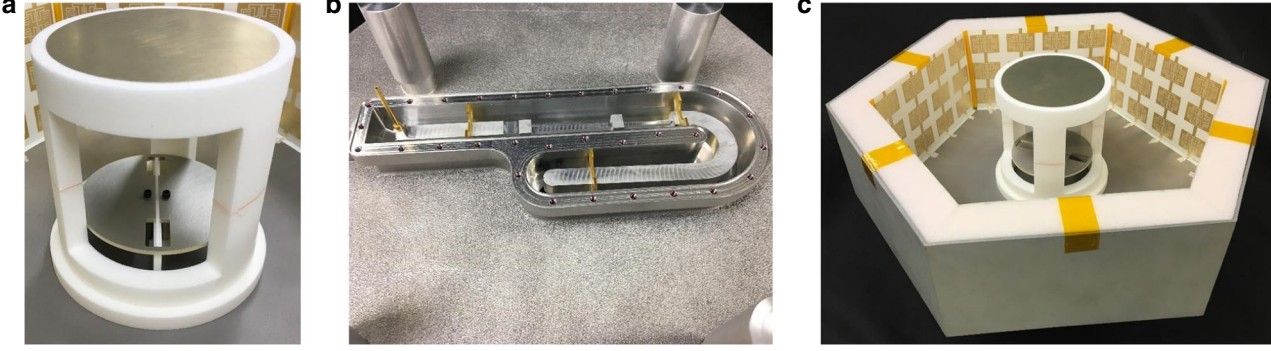

**Fig. 3** The fabricated A-SBFA. **a** Fabricated slotted patch feed and subreflector with 3D-printed mounting fixture. Both were machined out of aluminum. **b** Underside of A-SBFA, showing fabricated impedance matching network. **c** View of assembled A-SBFA showing 3D-printed aluminum cavity, feed antenna and subreflector, as well as copper metasurfaces etched on a thin substrate and separated from cavity walls by a low permittivity foam layer

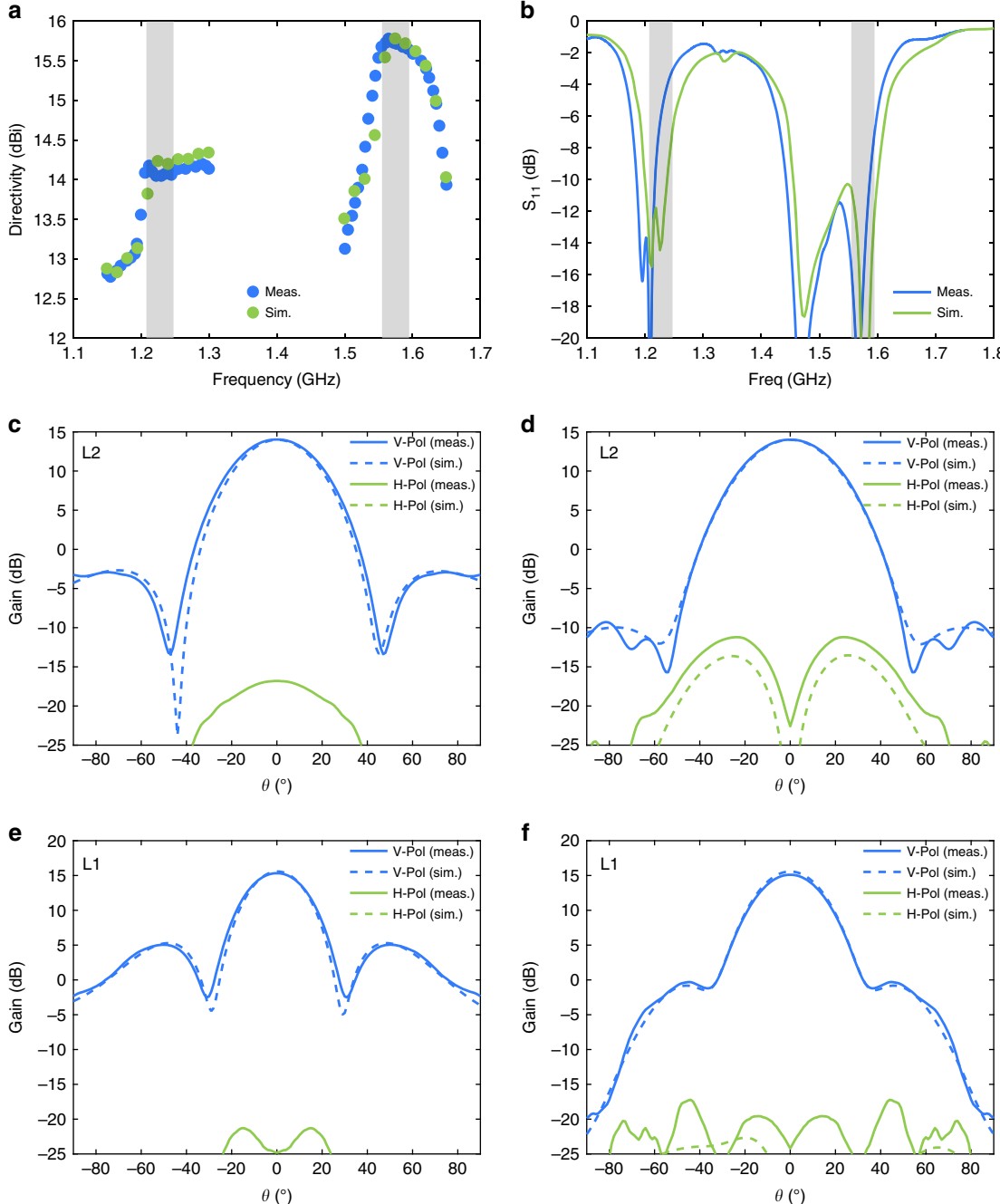

**Fig. 4** Simulated and measured A-SBFA performance. **a** Comparison of measured and simulated directivity versus frequency with frequency bands of interest indicated. **b** Simulated and measured $S_{11}$ of A-SBFA with impedance matching network. **c** Comparison of simulated and measured A-SBFA gain at L2 (E-plane cut). Both the co-polarized (V-pol) and cross-polarized (H-pol) gain patterns are shown. The E-plane is the $y$–$z$ plane in Fig. 1a. **d** Comparison of simulated and measured A-SBFA gain at L2 (H-plane cut). The H-plane is the $x$–$z$ plane in Fig. 1a. **e** Comparison of simulated and measured A-SBFA gain at L1 (E-plane cut). **f** Comparison of simulated and measured A-SBFA gain at L1 (H-plane cut)

cable, and was also machined out of aluminum. This network consisted of several series stubs, whose length and diameter were optimized to provide a good impedance match at L1 and L2. These stubs enabled the transformation between the 50 Ω input and the antenna input impedance. The sub-reflector was held in place above the slot-loaded patch by a lightweight foam support structure. The complete manufactured antenna is shown in Fig. 3c.

**Measured results**. Measured directivity as a function of frequency is plotted in Fig. 4a. Directivity is relatively stable within each

band, and peak directivities are 14.1 dBi and 15.7 dBi at L2 and L1, respectively, corresponding to aperture efficiencies of 97% at L2 and 85% at L1. The simulated and measured $S_{11}$, shown in Fig. 4b, agree very well with each other, except for a slight downward frequency shift of around 15 MHz. The frequency shift is likely due to minor differences between the modeled and manufactured matching networks, and a slight re-design of the matching network based on the measured antenna output impedance is expected to remedy this. The far-field directivity patterns, shown in Fig. 4c–f, also agree very well with simulated predictions. Material losses due to the copper patterning and

dielectric substrate that comprise the metasurface contribute to a difference of 0.1 dB predicted between peak directivity and peak gain. Total losses measured by the Wheeler cap method[25] are 0.2 dB. The directivity patterns demonstrate the outstanding polarization integrity of this antenna, since the peak cross-polarization is more than 25 dB less than the peak co-polarization. This is a result of the nearly identical E and H-plane patterns due to an almost uniform aperture excitation. For applications where circular polarization is required a mean-der line polarizer will be employed[26], or the A-SBFA can be fed using a circularly polarized feed such as a crossed-slot patch or a crossed dipole[27,28]. When the A-SBFA is used in high-power applications, corona and multipaction are of concern near the feed where the electric field magnitude is most intense. For this reason a dielectric foam plug may be inserted from the circular opening in the ground plane, along the parallel wire transmission line and in the subreflector gap.

To further illustrate the flexibility and power of this design procedure, we performed an additional optimization to extend the high aperture efficiency condition to a third GPS frequency band, L5, which is around 1.176 GHz. The optimized surface impedance properties of the metasurface required to achieve this result are shown in Supplementary Fig. 12 and the corresponding simulated far field performance is shown in Supplementary Fig. 13. Simulated aperture efficiency is 95.6% at L5, compared to around 80%, which was the aperture efficiency in the original design. Moreover, the L2 and L1 aperture efficiencies remain high at 93.5% and 86.5%, respectively. Although these results are theoretical and are yet to be demonstrated experimentally, they are further evidence that the addition of metasurfaces to the hexagonal SBFA provides unprecedented flexibility for achieving high aperture efficiency at multiple frequency bands.

## Discussion

The results shown here serve to validate the design as an exceptional candidate for future satellite-based communications systems. The A-SBFA achieves a dual-band capability, high aperture efficiency, and excellent cross-polarization while still maintaining a light weight and low height profile. Whereas the hexagonal SBFA without metasurfaces has an average aperture efficiency of 68% at the L1 and L2 bands, our antenna achieves an average aperture efficiency of over 90%. This corresponds to a directivity increase of approximately 1.5 dB at L2 and 0.9 dB at L1. Thus, to achieve a given gain requirement the array can be made smaller, leading to significant size, weight and power sav-ings. This is accomplished without the use of dielectric slabs, significantly reducing weight and making this antenna highly desirable for space applications. Furthermore, the A-SBFA we present here offers a factor of five height reduction compared with the currently used helical GPS antennas. Additionally, the hexagonal shape of this A-SBFA allows it to be efficiently deployed in an array environment. During the antenna design process, not only was the electromagnetic radiation performance optimized, but the A-SBFA was also simultaneously engineered to meet the harsh environmental conditions of space, with the 3D printed cavity and grounded metasurface design resulting in low multipaction, PIM, and ESD risk, making it well suited to high power operation while in a low pressure or vacuum environment. Furthermore, these results provide a compelling demonstration of the value of metamaterials, in that they may be used to augment conventional antenna technology and enable performance that would otherwise be impossible for state-of-the-art technology.

## Methods

**Simulation methodology.** Antenna full-wave simulations were performed using the commercially available Ansys HFSS and CST Microwave Studio codes. In the

initial simulations, the metasurfaces were modeled using anisotropic impedance surfaces to reduce the computational burden during the design process, but the actual metasurface structure was included in the final full-wave simulations.

**Measurement methodology.** Far-field patterns for the A-SBFA were measured in a tapered anechoic chamber from 1100–1750 MHz. A diagram of the antenna test range is shown in Supplementary Fig. 14, as well as a photograph of the A-SBFA demonstration unit mounted on the range positioner. The antenna was char-acterized in receive mode, where a transmitted signal swept from 1100–1750 MHz collimates into a plane wave that impinges upon the A-SBFA. The two-axis positioner on top of the antenna pedestal enabled the collection of far field energy in great circle cuts about the forward hemisphere of the A-SBFA. The range of far field points in $\theta$ was 0° to 90° with a step of 1°, and a range in $\varphi$ of −180° to 180° at 15° steps. The far field gain was measured by comparing the received signal to that of a SA12-1.1 pyramidal standard gain horn. Antenna efficiency was measured using the Wheeler cap method[25]. An Agilent E8362C PNA functioned as the receiver, and the data was processed using the OrbitFR v6.0 software and custom Matlab scripts. The antenna directivity was calculated by numerically integrating the measured fields.

## Data availability

The data that support the findings of this study are available from the corre-sponding author on request.

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

## Acknowledgements

The authors would like to thank Micah Gregory for his assistance in the design of the matching network. This work was sponsored by Lockheed Martin Corporation, contract U16-002.

## Author contributions

E.L. conceived the idea, D.H.W. and Z.H.J. developed the design methodology, J.D.B. carried out the design and optimization of the device, T.H. oversaw manufacture of the device and helped design the matching network, D.H.W. oversaw the project. All the authors contributed to the interpretation of results and participated in the preparation of the manuscript.

## Additional information

**Competing interests:** The authors declare no competing interests.

