## [Peer Review File · Nature Communications]

Reviewers' Comments:

Reviewer #1:

Remarks to the Author:

Paper presents the utilization of a metamaterial surface to enhance the performances of a relatively compact antenna on two bands.

Paper is clear, complete for understanding and concise.

Authors claims to have designed a high efficiency compact SBFA antenna based on metamaterials. They provide references to relevant papers in the field to allow comparison and verification of their superior results. Results are interesting for the EM community.

As minor notes:

- The basic cell of the metasurface is shown without dimensions the same is true for the antenna, so results are not reproducible. Authors should present the layout of the cell and of the antenna with all lengths and widths in nthe supplementary material

- Authors might wish to extend the lower limit in Fig.2 subfigures c,d,e,f to -25dB to let the reader better appreciate the cross-polar curves, which are completely absent in subfigure e and hardly visible in subfigure f.

Reviewer #3:

Remarks to the Author:

This paper describes modifications to the classical short backfire antenna (SBFA), primarily to achieve better aperture efficiency and dual-band operation. The key modification proposed is to line the metallic walls of the SBFA with a printed metasurface which acts as a dual band 'hard wall' thus increasing aperture efficiency. The overall structure has been manufactured and experimentally tested for a GPS application.

Although the work is quite interesting and potentially useful, most of the proposed concepts have been reported before by the authors and others. For example, reference [17] introduced the idea of using metamaterial linings to horn antennas to improve their radiation pattern characteristics. Primarily [17] deals with 'soft' walls but the key concept used, i.e. exciting hybrid modes by means of printed anisotropic impedance walls is similar. Likewise, the key ideas presented in the paper under review have been published before by the authors in reference [7]. Namely, the use of the same dual-band 'hard' walls to synthesize an improved SBFA (albeit with circular cross section, although the hexagonal cavity is also mentioned in [7]).

Some specific comments are described below:

1. In the Introduction it is stated that the aperture efficiency of the standard SBFA is 84%. Yet in the proposed lined version, the efficiency at L1 is reported as 85% . Only the efficiency at L2 is improved. Isn't this a drawback of this design? To be successful, shouldn't the aperture efficiency be improved at both L1 and L2?
2. In the introduction, it is stated that these hexagonal apertures are well suited for electronic beam steering. But since they have a width of 2 wavelengths at L1 won't they introduce a problem with grating lobes? And how important is to have high-aperture efficiency elements when they are restricted close to a half wavelength width in a scanning application?
3. How were the anisotropic impedances at L1 and L2 chosen? Is the TE/TM impedance balanced condition (page 8 of [17]) relevant here? Was this condition met?
4. In Figures 3, 4 which is the E and which is the H plane exactly?
5. I found it confusing to relate the patterns shown in Fig. 4 to the reported aperture efficiencies.

Figures 4c and 4d show the E and H plane at L2 where the aperture efficiency is reported to be 97%. Nevertheless, the E-plane sidelobes seem close to -17dB whereas the H-plane sidelobes seem close to -25dB. These sidelobe levels closely match those of the TE₁₁ mode on a circular aperture with only 84% aperture efficiency.

6. Perhaps the high aperture efficiency is achieved at L1 (Figures 4e and 4f; reported aperture efficiency 85%) where the sidelobes are higher in the H plane and lower cross-polarization is achieved.

7. Because this is a hexagonal aperture and there is no standard theory about it, in the supplementary, the E,H patterns of a commensurate unlined hexagonal SBFA should be directly compared to the lined ones.

8. If this antenna is meant for GPS how is circular polarization obtained?

9. Aren't the fine gaps on the dual-band patches shown in Figure 2 susceptible to arcing at higher power levels?

Overall this is solid work but I am not convinced that there is sufficient new information reported here to justify publication in Nature Communications (and the rather standard fabrication and testing of the antenna is not sufficient in my opinion).

Reviewer #4:

Remarks to the Author:

The paper presents the realization and experimental validation of an innovative antenna particularly suitable for space applications. The design is based on the idea presented in ref. [7], which suggested the use of hard boundary conditions on the inside walls of the main reflector to increase the directivity of a short backfire antenna. The present paper generalizes this concept by optimizing the anisotropic boundary conditions and verifies the design through fabrication and experimental characterization. Furthermore, the solution is engineered to make it suitable for space applications by reducing the weight through the use of foam instead of a dielectric slab and by making the antenna hexagonal for a more efficient packaging. Also, special measures are taken to reduce the risk of electrostatic discharge, passive intermodulation and multipaction.

The paper is very interesting, and generally clear and well written. The results appear novel and of interest from an applicative point of view.

Just a few comments:

- Sometimes, the text could be rearranged to make it easier to follow. For instance, the description of metasurfaces (page 4) and of their microwave implementation (page 6) could be anticipated;
- In page 3, when presenting the content of the work, the authors talk about performance achieved by using metasurfaces that have been optimized to enhance directivity at two frequency bands or "over a wide band". However, it seems that presented results are only relevant to the two-band operation;
- The paper proposes metasurface lining as a generalization of the hard boundary condition ("In contrast to conventional hard surfaces, metasurfaces exhibit a wider range of anisotropic and dispersive properties"), however the authors in the following talk about a "hard metasurface" (page 4). Looking at the impedance values at L2, the metasurface does not seem to impose hard boundary conditions;
- Is the surface reactance extracted from simulated scattering parameters independent from the incidence direction? If not, how is chosen the incidence direction to be used?

PennState

Douglas H. Werner

*John L. and Genevieve H. McCain Chair Professor, Department of Electrical Engineering
Director, Computational Electromagnetics and Antennas Research Lab (CEARL)*

October 30, 2018

Dear *Nature Communications* Editors and Reviewers,

We would like to thank the reviewers for their helpful suggestions in improving our manuscript. Our responses to specific reviewer comments are provided below.

Reviewer #1 (Remarks to the Author):

Paper presents the utilization of a metamaterial surface to enhance the performances of a relatively compact antenna on two bands.

Paper is clear, complete for understanding and concise.

Authors' claim to have designed a high efficiency compact SBFA antenna based on metamaterials. They provide references to relevant papers in the field to allow comparison and verification of their superior results. Results are interesting for the EM community.

As minor notes:

Reviewer #1, comment 1:

The basic cell of the metasurface is shown without dimensions the same is true for the antenna, so results are not reproducible. Authors should present the layout of the cell and of the antenna with all lengths and widths in the supplementary material

Response to Reviewer #1, comment 1:

We thank the reviewer for bringing our attention to this oversight. Detailed dimensions of the antenna and metasurface are included in Supplementary Figs. 1 and 8 and Supplementary Tables 1 and 3.

Supplementary Figure 1. Dimensions of A-SBFA feed antenna and subreflector.
Sizes of each dimension are listed in Supplementary Table 1.

Supplementary Figure 8. Dimensions of metasurface unit cell.
 Sizes of each dimension are listed in Supplementary Table 3.

Supplementary Table 1. Sizes of SBFA dimensions in Supplementary Figure 3, given in terms of the wavelength at 1.4 GHz.

Dimension	Size (λ)
a	0.1551
b	0.0314
c	0.0097
d	0.0593
e	0.0994
f	0.2058
g	0.4777
h	0.4144
i	0.0962

Supplementary Table 3. Sizes of metasurface dimensions in Supplementary Figure 8, given in terms of the wavelength at 1.4 GHz.

Dimension	Size (λ)
h_1	0.0303
h_2	0.1104
h_3	0.0133
h_4	0.0456
h_5	0.0286
h_6	0.0657
w_1	0.0236
w_2	0.0400
w_3	0.0221
w_4	0.0108
w_5	0.0641
w_6	0.0243
w_7	0.0195
w_8	0.0959
t_1	0.0011
t_2	0.0028
t_3	0.0024
t_4	0.0033
t_5	0.0028

Reviewer #1, comment 2:

Authors might wish to extend the lower limit in Fig.2 subfigures c,d,e,f to -25dB to let the reader better appreciate the cross-polar curves, which are completely absent in subfigure e and hardly visible in subfigure f.

Response to Reviewer #1, comment 2:

Figures 6c-f have been modified accordingly so that the cross-polarization curves are more visible.

Figure 4. Simulated and measured A-SBFA performance

a. Comparison of measured and simulated directivity versus frequency with frequency bands of interest indicated. **b.** Simulated and measured S_{11} of A-SBFA with impedance matching network. **c.** Comparison of simulated and measured A-SBFA gain at L2 (E-plane cut). Both the co-polarized (V-pol) and cross-polarized (H-pol) gain patterns are shown. The E-plane is the y - z plane in Fig. 1a. **d.** Comparison of simulated and measured A-SBFA gain at L2 (H-plane cut). The H-plane is the x - z plane in Fig. 1a. **e.** Comparison of simulated and measured A-SBFA gain at L1 (E-plane cut). **f.** Comparison of simulated and measured A-SBFA gain at L1 (H-plane cut).

Reviewer #2 (Remarks to the Author):

This paper describes modifications to the classical short backfire antenna (SBFA), primarily to achieve better aperture efficiency and dual-band operation. The key modification proposed is to line the metallic walls of the SBFA with a printed metasurface which acts as a dual band ‘hard wall’ thus increasing aperture efficiency. The overall structure has been manufactured and experimentally tested for a GPS application.

Although the work is quite interesting and potentially useful, most of the proposed concepts have been reported before by the authors and others. For example, reference [17] introduced the idea of using metamaterial linings to horn antennas to improve their radiation pattern characteristics. Primarily [17] deals with ‘soft’ walls but the key concept used, i.e. exciting hybrid modes by means of printed anisotropic impedance walls is similar. Likewise, the key ideas presented in the paper under review have been published before by the authors in reference [7]. Namely, the use of the same dual-band ‘hard’ walls to synthesize an improved SBFA (albeit with circular cross section, although the hexagonal cavity is also mentioned in [7]).

Some specific comments are described below:

Reviewer #2, comment 1:

In the Introduction it is stated that the aperture efficiency of the standard SBFA is 84%. Yet in the proposed lined version, the efficiency at L1 is reported as 85% . Only the efficiency at L2 is improved. Isn’t this a drawback of this design? To be successful, shouldn’t the aperture efficiency be improved at both L1 and L2?

Response to Reviewer #2, comment 1

Thank you for drawing our attention to this issue. We modified the introduction to improve clarity and underscore the fact that without the use of metasurfaces, high average aperture efficiency is not achievable across multiple discrete frequency bands for the hexagonal SBFA. The aperture efficiency of a standard circular SBFA is indeed around 84%, but this is over a narrow bandwidth. The standard SBFA cannot simultaneously achieve high aperture efficiency across two frequency bands. Furthermore, the aperture efficiency of a hexagonal SBFA is slightly lower than that of the circular SBFA. To demonstrate this we optimized a hexagonal SBFA without metasurfaces and found that the highest aperture efficiency that could be simultaneously achieved at L1 and L2 was approximately 68.5% and 67.2% respectively. Additionally, the highest aperture efficiency we could achieve at a single frequency was less than 75%. Please see Supplementary Figs. 3 and 4 for more information. We have also found that hexagonal SBFAs have significantly improved cross-polarization compared with conventional circular SBFAs due to E- and H-plane asymmetry, and even more improved compared to conventional hexagonal SBFAs. This problem is greatly reduced in the optimized A-SBFA. The text we added to the introduction is included below:

We have found that the maximum aperture efficiency achievable at a single frequency for a hexagonal SBFA is below 75%, and the highest average aperture efficiency simultaneously achievable at two frequencies is around 68%.

Additionally, we refer to these figures in the second paragraph of the Results section.

Supplementary Figure 3. Dimensions of SBF optimized with no metasurface.

Supplementary Figure 4. Simulated directivity of optimized SBFA without metasurfaces.

The dimensions of an SBFA without metasurfaces were optimized to maximize directivity at L1 and L2. A global optimization algorithm was run for several thousand iterations to achieve these results. Simulated directivity of this optimized antenna is shown here. The H-plane is the x - z plane in Supplementary Fig. 3. The E-plane is the y - z plane in Supplementary Fig. 3. **a.** Peak L2 directivity is 12.5 dBi, corresponding to an aperture efficiency of 67.2%. **b.** Peak L1 directivity is 14.8 dBi, corresponding to an aperture efficiency of 68.5%.

Supplementary Table 2. Sizes of SBFA dimensions in Supplementary Figure 3, given in terms of the wavelength at 1.4 GHz.

Dimension	Size (λ)
a	0.1422
b	0.0314
c	0.0095

d	0.0711
e	0.1070
f	0.2151
g	0.4556
h	0.4133
i	0.0879
j	0.4827
k	0.8890

Reviewer #2, comment 2:

In the introduction, it is stated that these hexagonal apertures are well suited for electronic beam steering. But since they have a width of 2 wavelengths at L1 won't they introduce a problem with grating lobes? And how important is to have high-aperture efficiency elements when they are restricted close to a half wavelength width in a scanning application?

Response to Reviewer #2, comment 2:

The statement about electronic beam steering refers to applications with approximately ± 15 deg field of view. Therefore, this antenna is ideal in a GPS application with a ± 14 deg FOV or other MEO (medium earth orbit) applications, (e.g. TDRSS satellites where short backfire antennas have been used in the past [22]). Please see Supplementary Fig. 2 for more information.

The short backfire antenna performs best when its diameter is approximately 1.7-2.2 wavelengths, so it cannot be used for wide scan array applications that require small inter-element spacing. We added the following text to the first paragraph of the Results section to clarify these important points:

An array with a 2λ element spacing corresponds to a $\pm 14^\circ$ field of view, which is ideal for medium earth orbit (MEO) applications such as GPS (Supplementary Fig. 2).

22. Nessel, J. A., Kory, C. L., Lambert, K. M., Acosta, R. J. & Miranda, F. A. A microstrip patch-fed short backfire antenna for the tracking and data relay satellite system-continuation (TDRSS-C) multiple access (MA) array. In *2006 IEEE Antennas and Propagation Society International Symposium*, 521–524 (2006).

Supplementary Figure 2. Theoretical scanning performance of SBFA array with 2λ element spacing.

Medium Earth Orbit (MEO) satellite applications such as GPS require an antenna array with a field of view (FOV) of around $\pm 14^\circ$. The above figure demonstrates that in the case of an array with 2λ element spacing, the grating lobes fall outside of the field of view when scanning to 14° radially at both L1 and L2.

Reviewer #2, comment 3:

How were the anisotropic impedances at L1 and L2 chosen? Is the TE/TM impedance balanced condition (page 8 of [17]) relevant here? Was this condition met?

Response to Reviewer #2, comment 3:

The anisotropic impedances at L1 and L2 were chosen through a global optimization algorithm. We did not limit the optimization to just the values that meet this balanced condition. However, the optimized case does approximate a hard surface which was the basic assumed starting point for the A-SBFA design from the beginning. This proves that the design concept and the optimization are consistent. See also the response to Reviewer #3, comment 3. The following text was added to the Results section to clarify this point:

Rather than limit the optimization to only choose surface reactance values that would approximate a hard boundary or fulfill the balanced hybrid condition¹⁷ we allowed the algorithm to explore the entire design space.

Reviewer #2, comment 4:

In Figures 3, 4 which is the E and which is the H plane exactly?

Response to Reviewer #2, comment 4:

Fig. 1a has been modified to include axes, which are then referenced in the Fig. 4 caption to help clarify this.

Figure 1. Short backfire antenna geometry

a. Model of hexagonal metamaterial-enabled A-SBFA. **b.** Visualization of size comparison between A-SBFA and helical antenna currently used aboard GPS satellite (complete helical array shown in inset).

Text added to Fig. 4 caption:

The E-plane is the y-z plane in Fig. 1a. The H-plane is the x-z plane in Fig. 1a.

Reviewer #2, comment 5:

I found it confusing to relate the patterns shown in Fig. 4 to the reported aperture efficiencies. Figures 4c and 4d show the E and H plane at L2 where the aperture efficiency is reported to be 97%. Nevertheless, the E-plane sidelobes seem close to -17dB whereas the H-plane sidelobes seem close to -25dB. These sidelobe levels closely match those of the TE₁₁ mode on a circular aperture with only 84% aperture efficiency.

Response to Reviewer #2, comment 5:

The aperture efficiency values given in this paper are calculated using the standard textbook formula involving peak directivity and physical aperture size. There are a couple of key differences between this antenna and conventional aperture antennas where the relationship between directivity, aperture efficiency, beamwidth and sidelobes are known from textbooks. One difference is that our antenna has a hexagonal aperture shape which results in higher order modes in the phi-direction. The main difference, however, is that this antenna radiates from an annular aperture as seen in the added aperture distribution plots (Supplementary Fig. 7). Therefore, the conventional relationship between aperture efficiency and sidelobes for a circular aperture cannot be expected to be very accurate in this case.

Supplementary Figure 7. Electric field magnitude cuts at L1 and L2.

The above plots show the magnitude of the electric fields 8 mm above the subreflector given an input power of 1 W. Dashed lines indicate the edge of the SBF aperture. **a.** Electric field magnitude at L2 in the H-plane. **b.** Electric field magnitude at L2 in the E-plane. **c.** Electric field magnitude at L1 in the H-plane. **d.** Electric field magnitude at L1 in the E-plane.

Reviewer #2, comment 6:

Perhaps the high aperture efficiency is achieved at L1 (Figures 4e and 4f; reported aperture efficiency 85%) where the sidelobes are higher in the H plane and lower cross-polarization is achieved.

Response to Reviewer #2, comment 6:

Please see our response to comment 5 above.

Reviewer #2, comment 7:

Because this is a hexagonal aperture and there is no standard theory about it, in the supplementary, the E, H patterns of a commensurate unlined hexagonal SBFA should be directly compared to the lined ones.

Response to Reviewer #2, comment 7:

Thank you for drawing our attention to this omission; this is an excellent suggestion. We have included E and H plane patterns of an optimized unlined hexagonal SBFA (Supplementary Fig. 4). Additionally, the following text was added to the Introduction:

We have found that the maximum aperture efficiency achievable at a single frequency for a hexagonal SBFA is below 75%, and the highest average aperture efficiency simultaneously achievable at two frequencies is around 68%.

Supplementary Figure 4. Simulated directivity of optimized SBFA without metasurfaces.

The dimensions of an SBFA without metasurfaces were optimized to maximize directivity at L1 and L2. A global optimization algorithm was run for several thousand iterations to achieve these results. Simulated directivity of this optimized antenna is shown here. The H-plane is the x - z plane in Supplementary Fig. 3. The E-plane is the y - z plane in Supplementary Fig. 3. **a.** Peak L2 directivity is 12.5 dBi, corresponding to an aperture efficiency of 67.2%. **b.** Peak L1 directivity is 14.8 dBi, corresponding to an aperture efficiency of 68.5%.

Reviewer #2, comment 8:

If this antenna is meant for GPS how is circular polarization obtained?

Response to Reviewer #2, comment 8:

Circular polarization can be achieved by including a multi-layer meanderline polarizer, which is located in the antenna aperture.

Additionally, this antenna could also be designed in the future to have a circularly polarized feed such as a crossed slot patch or crossed dipole [26]-[28]. To clarify this, the following text was added to the last paragraph in the Results section:

For applications where circular polarization is required a meander line polarizer could be employed²⁶, or the A-SBFA can be fed using a circularly polarized feed such as a crossed-slot patch or a crossed dipole²⁷⁻²⁸

26. Young, L., Robinson, L. & Hacking, C. Meander-line polarizer. *IEEE Transactions on Antennas and Propagation* **21**, 376–378 (1973).
27. Iwasaki, H. A circularly polarized small-size microstrip antenna with a cross slot. *IEEE Transactions on Antennas and Propagation* **44**, 1399–1401 (1996).
28. Nasimuddin, Esselle, K. P. & Verma, A. K. Study of various slots in circular patch for circularly polarized antennas and enhancing their gain by short horns. In *2006 Asia-Pacific Microwave Conference*, 1519–1522 (2006).

Reviewer #2, comment 9:

Aren't the fine gaps on the dual-band patches shown in Figure 2 susceptible to arcing at higher power levels?

Response to Reviewer #2, comment 9:

The power level is most intense at the feed, so corona and multipaction are of concern. However, for high power applications a dielectric foam plug may be inserted from the circular opening in the ground plane, along the parallel wire transmission line and in the subreflector gap. The electric field intensity at the metasurface is not as intense since the power from the feed is distributed over the entire surface area of the hexagonal aperture, resulting in substantially lower power density at the metasurface than within the feed. To clarify this, the following text was added to the Results section:

When the A-SBFA is used in high-power applications, corona and multipaction are of concern near the feed where the electric field magnitude is most intense. For this reason a dielectric foam plug may be inserted from the circular opening in the ground plane, along the parallel wire transmission line and in the subreflector gap.

Reviewer #2, comment 10:

Overall this is solid work but I am not convinced that there is sufficient new information reported here to justify publication in Nature Communications (and the rather standard fabrication and testing of the antenna is not sufficient in my opinion).

Response to Reviewer #2's assessment of this work:

We would like to thank the reviewer for drawing our attention to several areas where the paper could be improved and clarified. We also apologize for not making the major contributions of this work sufficiently clear in the manuscript text. The novelty aspect of this work is explained below:

While a circular SBFA with conventional hard surfaces can achieve high aperture efficiency, the addition of thick dielectric slabs is estimated to double the weight of the antenna. Furthermore, a circular aperture is not ideal for array applications due to the lower packaging efficiency compared to a triangular or hexagonal aperture. In this work we design a hexagonal A-SBFA with metasurfaces that allow for very high aperture efficiency across two discrete frequency bands. Whereas the hexagonal SBFA without metasurfaces has an average aperture efficiency of 68% at the L1 and L2 bands, our antenna achieves an average aperture efficiency of close to 90%. This corresponds to a directivity increase of approximately 1.5 dB at L2 and 0.9 dB at L1. Thus, to achieve a given gain requirement the array can be made smaller, leading to significant size, weight and power savings. This is achieved without the use of dielectric slabs, significantly reducing weight and making this antenna highly desirable for space applications. Furthermore, the A-SBFA we present here offers a factor of 5 height reduction compared with the currently used GPS antennas (see Figure 1 for a relative size comparison of the two antennas). This volume savings is critical for satellite applications. We believe the combined benefits of weight and volume savings along with superior experimentally demonstrated aperture efficiency performance represent a significant step forward for the space communications field. As an additional note we want to mention that the A-SBFA has been engineered to meet the harsh environmental space

conditions with a more controlled and robust design compared to the helical antenna, in particular with respect to multipaction. We have modified the manuscript in several places to help clarify the significance of this work. In particular we modified the Discussion section to read as follows:

Whereas the hexagonal SBFA without metasurfaces has an average aperture efficiency of 68% at the L1 and L2 bands, our antenna achieves an average aperture efficiency of over 90%. This corresponds to a directivity increase of approximately 1.5 dB at L2 and 0.9 dB at L1. Thus, to achieve a given gain requirement the array can be made smaller, leading to significant size, weight and power savings. This is accomplished without the use of dielectric slabs, significantly reducing weight and making this antenna highly desirable for space applications. Furthermore, the A-SBFA we present here offers a factor of 5 height reduction compared with the currently used helical GPS antennas. Additionally, the hexagonal shape of this A-SBFA allows it to be efficiently deployed in an array environment. During the antenna design process, not only was the electromagnetic radiation performance optimized, but the A-SBFA was also simultaneously engineered to meet the harsh environmental conditions of space, with the 3D printed cavity and grounded metasurface design resulting in low multipaction, PIM, and ESD risk, making it well suited to high power operation while in a low pressure or vacuum environment.

Reviewer #3 (Remarks to the Author):

The paper presents the realization and experimental validation of an innovative antenna particularly suitable for space applications. The design is based on the idea presented in ref. [7], which suggested the use of hard boundary conditions on the inside walls of the main reflector to increase the directivity of a short backfire antenna. The present paper generalizes this concept by optimizing the anisotropic boundary conditions and verifies the design through fabrication and experimental characterization. Furthermore, the solution is engineered to make it suitable for space applications by reducing the weight through the use of foam instead of a dielectric slab and by making the antenna hexagonal for a more efficient packaging. Also, special measures are taken to reduce the risk of electrostatic discharge, passive intermodulation and multipaction.

The paper is very interesting, and generally clear and well written. The results appear novel and of interest from an applicative point of view.

Just a few comments:

Reviewer #3, comment 1:

Sometimes, the text could be rearranged to make it easier to follow. For instance, the description of metasurfaces (page 4) and of their microwave implementation (page 6) could be anticipated;

Response to Reviewer #3, comment 1:

We thank the reviewer for pointing this out. In the course of revising the manuscript according to the reviewers' recommendations we made several changes that we hope will improve clarity and make the manuscript easier to follow.

Reviewer #3, comment 2:

In page 3, when presenting the content of the work, the authors talk about performance achieved by using metasurfaces that have been optimized to enhance directivity at two frequency bands or "over a wide band". However, it seems that presented results are only relevant to the two-band operation;

Response to Reviewer #3, comment 2:

In this work our design is targeting a dual-band application. We have not pursued wide band designs yet but we believe this design approach introduced in the paper can easily be extended to wide band designs. However, since a wide band design is beyond the scope of this work we have changed the text to:

This unprecedented performance is achieved through the use of interior foam walls lined with anisotropic and dispersive metasurfaces that have been optimized to enhance antenna directivity at two frequency bands.

Moreover, to further illustrate the flexibility and power of this design procedure, we performed an additional optimization to extend the high aperture efficiency condition to a third GPS frequency band, L5, which is around 1.176 GHz. The optimized surface impedance properties of the metasurface required to achieve this result are shown in Supplementary Fig. 12 and the corresponding simulated far field performance is shown in Supplementary Fig. 13. The following text was added to the end of the Results section:

To further illustrate the flexibility and power of this design procedure, we performed an additional optimization to extend the high aperture efficiency condition to a third GPS frequency band, L5, which is around 1.176 GHz. The optimized surface impedance properties of the metasurface required to achieve this result are shown in Supplementary Fig. 12 and the corresponding simulated far field performance is shown in Supplementary Fig. 13. Simulated aperture efficiency is 95.6% at L5, compared to around 80%, which was the aperture efficiency in the original design. Moreover, the L2 and L1 aperture efficiencies remain high at 93.5% and 86.5%, respectively. Although these results are theoretical and are yet to be demonstrated experimentally, they are further evidence that the addition of metasurfaces to the hexagonal A-SBFA provides unprecedented flexibility for achieving high aperture efficiency at multiple frequency bands.

Supplementary Figure 12. Theoretical anisotropic surface reactance values of metasurface required for optimized multi-band A-SBFA.

By increasing the parallel inductance of the unit cell we can shift the X_{xx} resonance to a lower frequency. Similarly, we can increase the series capacitance to decrease the magnitude of X_{yy} to fulfill the impedance surface requirements of the optimized A-SBFA design.

Supplementary Figure 13. Directivity and aperture efficiency of optimized multi-band A-SBFA.

Using new surface impedance parameters (Supplementary Fig. 12) the simulated directivity (a.) and aperture efficiency (b.) at L5 (1.176 GHz) are much improved. Simulated aperture efficiency is 95.6% at L5, 93.5% at L2, and 86.5% at L1.

Reviewer #3, comment 3:

The paper proposes metasurface lining as a generalization of the hard boundary condition (“In contrast to conventional hard surfaces, metasurfaces exhibit a wider range of anisotropic and dispersive properties”), however the authors in the following talk about a “hard metasurface” (page 4). Looking at the impedance values at L2, the metasurface does not seem to impose hard boundary conditions;

Response to Reviewer #3, comment 3:

We apologize for not clarifying this sufficiently in the text. The effective surface impedance properties shown in Fig. 2 are represented by Z_s in Supplementary Fig. 9. This should not be confused with Z_{in} , which is calculated from the reflection coefficient that includes the effects of the metallic ground plane representing the A-SBFA cavity walls. An antenna with ideal hard surfaces would fulfill the condition $|\text{Im}\{Z_{in,xx}\}| \rightarrow 0$, $|\text{Im}\{Z_{in,yy}\}| \rightarrow \infty$. From Supplementary Fig. 11 it can be seen that the conditions at L1 and L2 are approximately hard ($|\text{Im}\{Z_{in,xx}\}| \ll |\text{Im}\{Z_{in,yy}\}|$). The surface reactance values were arrived at through optimization and we did not wish to limit the optimization by imposing the strictly hard boundary condition case. Also see response to Reviewer #2, comment 3. The following text was added to the Results section to clarify this point:

Although a hard boundary condition was not imposed in the optimization, when considering the anisotropic effective input impedance of the metasurface placed above a ground plane representing the cavity walls, an approximately hard boundary condition is achieved at L1 and L2 (Supplementary Fig. 11).

We also added the following Supplementary Figure:

Supplementary Figure 11. Anisotropic effective input impedance of metasurface unit cell in a periodic environment, placed 33.6 mm above a ground plane.

When the metasurface unit cell is placed 33.6 mm above a ground plane (see Supplementary Fig. 9) the effective anisotropic input impedance can be calculated from the reflection coefficient. This input impedance is anisotropic due to the anisotropy of the metasurface. Note that the magnitude of the y -component of Z_{in} is much larger than the x -component near the L1 and L2 frequency bands, resulting in an approximately hard boundary condition ($|\text{Im}\{Z_{in,xx}\}| \ll (|\text{Im}\{Z_{in,yy}\}|)$).

Reviewer #3, comment 4:

Is the surface reactance extracted from simulated scattering parameters independent from the incidence direction? If not, how is chosen the incidence direction to be used?

Response to Reviewer #3, comment 4:

The A-SBFA optimization was performed using homogeneous anisotropic impedance surfaces that are angle invariant. When designing the actual metasurface we extracted the effective surface impedance parameters based on an assumed normally incident plane wave. It was found that although the effective surface impedance parameters vary slightly with incidence angle, the parameters extracted using normal incidence were sufficiently accurate for our application. We mentioned in the manuscript that when we initially simulated the A-SBFA with the metasurface the regions of high directivity were shifted to slightly different frequencies. Although we believe this is chiefly due to the truncation effect of using a finite metasurface it could also be caused by the angular dependence of the metasurface. Nevertheless, the problem was easily rectified by retuning one of the metasurface dimensions to shift the high directivity regions back to the L1 and L2 bands. After re-tuning the metasurface we observed very strong agreement between the performance of the homogeneous impedance surface A-SBFA and the metasurface A-SBFA, indicating that our extraction method is sufficiently accurate. There are methods of optimizing metasurfaces for robustness to angle of incidence so if we had found this to be a significant problem we could have modified the design process to account for this. The following text was added to the manuscript in the Results section:

The unit cell was illuminated with a normally incident plane wave and the effective surface reactance values of the metasurface were calculated from the simulated scattering parameters using transmission line theory (Fig. 2b. See also Supplementary Note 1 and Supplementary Fig. 9).

Additionally, since the effective surface reactance of the metasurface is weakly dependent on incidence angle, the surface impedance resonances can shift in frequency due to the direction of wave propagation within the cavity.

Sincerely,

Douglas H. Werner

Corresponding author:

Douglas H. Werner
Department of Electrical Engineering
The Pennsylvania State University
University Park, PA 16802
Phone: 814-863-2946
Email: dhw@psu.edu

Reviewers' Comments:

Reviewer #3:

Remarks to the Author:

I am satisfied with the technical responses to my comments.

Reviewer #4:

Remarks to the Author:

The authors have satisfactorily addressed all my comments